

# 1 Inferring the Photolysis Rate of NO₂ in the Stratosphere
# 2 Based on Satellite Observations

Jian Guan[1], Susan Solomon[1], Sasha Madronich[2,3], Douglas Kinnison[2]
[1]Department of Earth, Atmospheric, and Planetary Sciences, MIT; Cambridge, MA, 02139, USA
[2]Atmospheric Chemistry Observations and Modeling Laboratory, National Center for Atmospheric Research, Boulder,
CO, 80301, USA
[3]USDA UV-B Monitoring and Research Program, Natural Resource Ecology Laboratory Colorado State University, Fort
Collins, Colorado, 80523, USA
Correspondence to: Jian Guan (jianguan@mit.edu)
**Abstract.** NO and NO₂ (NOₓ) play major roles in both tropospheric and stratospheric chemistry. This paper provides a novel
method to obtain a global and accurate photodissociation coefficient for NO₂ based on satellite data. The photodissociation
coefficient $J_{NO_2}$ dominates the daytime diurnal variation of NOx photochemistry. Here the spatial variation of $J_{NO_2}$ in 50° S-90° S
in December from 20-40 km is obtained using data from the Michelson Interferometer for Passive Atmospheric Sounding (MIPAS)
experiment. Because NO and NO₂ exchange rapidly with one another in the daytime, the $J_{NO_2}$ can be attained assuming steady
state, and the results are shown to be consistent with model results. The $J_{NO_2}$ value decreases as the solar zenith angle increases
and has a weak altitude dependence. A key finding is that the satellite-derived $J_{NO_2}$ increases in the polar regions in good agreement
with model predictions, due to the effects of ice and snow on surface albedo. Thus, the method presented here provides an
observations-based check on the role of albedo in driving polar photochemistry.

## 19 1 Introduction

Fast photochemistry in the Earth's atmosphere is driven by sunlight and affects the diurnal variation of many species. The properties
of sunlight entering the stratosphere, including light intensity and its energy distribution, depend on the solar zenith angle, as well
as the overhead concentrations of oxygen and ozone and the reflective properties of the underlying troposphere. Further, the solar
zenith angle is related to latitude, season, and local time. The sunlight entering the stratosphere determines the photochemical rates
in the stratosphere, thus affecting stratospheric chemistry, and the diurnal variations of species concentration is one of the impacts.
Therefore, diurnal variation observations provide key information in analyzing the photochemical properties of the stratosphere.
NOx chemistry is one of the most important elements of stratospheric chemistry and plays a leading role in controlling stratospheric
ozone concentration (Crutzen, 1979; Johnston, 1971; Crutzen, 1970). The photodissociation coefficient $J_{NO_2}$ quantifies the process
of NO₂ photolysis into NO, thus affecting the diurnal variation of NOx. The stratospheric NO and NO₂ abundances are controlled
by the following reactions:
$NO + O_3 \rightarrow NO_2 + O_2$ (R1)
$NO_2 + h\nu \rightarrow NO + O$ (R2)
$NO_2 + O \rightarrow NO + O_2$ (R3)
$ClO + NO \rightarrow NO_2 + Cl$ (R4)
Because of the short lifetime of NO and NO₂, they are in steady state within the sunlit stratosphere. Therefore, the following
equation holds:



$$\frac{[NO]}{[NO_2]} \approx \frac{J_{NO_2} + k_{O+NO_2} \times [O]}{(k_{NO+O_3} \times [O_3] + k_{NO+ClO} \times [ClO])} \qquad (1)$$

A number of studies on the diurnal variation of NOx and $J_{NO_2}$ in the stratosphere have been reported, based on models or airborne
observations. Fabian et al. (1982) used a two-dimensional model to examine the diurnal variations of NOx at different altitudes.
Many studies of the NOx diurnal variation based on airborne observations which were then compared with models (Pommereau,
1982; Roscoe et al., 1986; Kawa et al., 1990). Madronich et al. (1985) measured $J_{NO_2}$ in the stratosphere utilizing a balloon platform
and compared it to a model; they showed that the $J_{NO_2}$ value has a weak altitude dependence. Webster and May (1987) measured
the diurnal variation of NOx and $J_{NO_2}$ simultaneously utilizing a balloon. Del Negro et al. (1999) calculated $J_{NO_2}$ based on the
concentrations of NO, $NO_2$, $O_3$, ClO, and $HO_2$ measured on an aircraft and BrO from a model, and compared them with a model.
They found that the $J_{NO_2}$ inferred from the data assuming steady state matched their model well. Moreover, it has been emphasized
that albedo has a substantial effect on $J_{NO_2}$ (Madronich, 1987; Bösch et al., 2001; Laepple, 2005; Walker et al., 2022). Further, the
surface albedo over ice and snow has a large and important effect on tropospheric chemistry in the polar regions (Walker et al.,
2022) due in large part to its effect on $J_{NO_2}$, highlighting the need to evaluate $J_{NO_2}$ on a large scale. Surface radiometers have also
been used to infer information about $J_{NO_2}$ for different sky conditions in the troposphere (Shetter et al., 1992; Junkermann et al.,
1989). However, aircraft, surface radiometers, or balloon measurements are all local and the amount of data is therefore limited.
At the same time, models are based on theoretical calculations and require measured data for verification. These considerations
are the motivation for this paper, in which satellite data are used to characterize $J_{NO_2}$ on a global basis, with particular emphasis on
values obtained over ice and snow.
Satellite measurements of NOx allow elucidation of its zenith angle and albedo dependence. The global concentrations of NO,
$NO_2$, and related species as discussed below can be easily obtained using satellite data and used to determine $J_{NO_2}$ at different
latitudes, albedo, and altitudes. Solomon et al. (1986) reported satellite observations of the $NO_2$ diurnal variation in the stratosphere
at solar zenith angles ranging from about 35 to 110 degrees but concurrent NO data were not available. Anderson et al. (1981)
employed a similar method to study the zenith angle variation of mesospheric $O_3$. The Michelson Interferometer for Passive
Atmospheric Sounding (MIPAS) is a backscatter Fourier transform spectrometer carried on Envisat, measuring not only $NO_2$ but
also NO and $O_3$, as well as ClO, all of which are used here in inferring $J_{NO_2}$ (see below). MIPAS was designed and operated for
the measurement of atmospheric species from space and can detect limb emission in the middle atmosphere with high spectral
resolution and low-noise performance (Fischer et al., 2008).
In this work, the novel method of obtaining the zenith angle dependence of NOx and $J_{NO_2}$ using satellite data in summer over the
polar cap is reported, taking 50° S-90° S in December in 20-40 km as an example. The diurnal variations of NOx and $J_{NO_2}$ at
different altitudes are described. $J_{NO_2}$ changes with latitude are discussed and a $J_{NO_2}$ map in the Antarctic is used to elucidate
albedo effects. In summary, this work shows a method for obtaining NOx diurnal variation and accurate $J_{NO_2}$ based on satellite
data, expanding the way to attain information on this key photodissociation coefficient.
**2    Data and Methods**
**2.1 MIPAS Data**
The vertical resolution of MIPAS is approximately 3 km and the horizontal resolution of MIPAS is 30 km, and the vertical scan
range is 5-150 km. Satellite operation was stopped temporarily in March 2004 due to technical issues and resumed in January 2005
in a new operation mode. MIPAS allows near complete global coverage, ranging from 87° S to 89° N obtained about every three



72 days by 73 scans per orbit and 14.3 orbits per day. Each day the satellite passes through the same latitude at two local times

73 (ascending side and descending side, as shown in Fig. 1). Therefore, for this dataset, there are only two solar zenith angles at each

74 latitude. We therefore focus on 50° S-90° S in the polar day, in December 2009, where there are as many solar zenith angles as

75 possible in a relatively small latitude range. In this paper, the NO, $NO_2$, $O_3$, ClO, temperature and pressure data were from V8 level

76 2 MIPAS retrievals (Kiefer et al., 2021, 2022). The reported precision between 20 km and 40 km is 5-15 % for NO (Funke et al.,

77 2022; Sheese et al., 2016), 5-15 % for $NO_2$ (Wetzel et al., 2007), 2-3 % for $O_3$ (Laeng et al., 2015) and more than 35 % for ClO

78 (Von Clarmann et al., 2009).

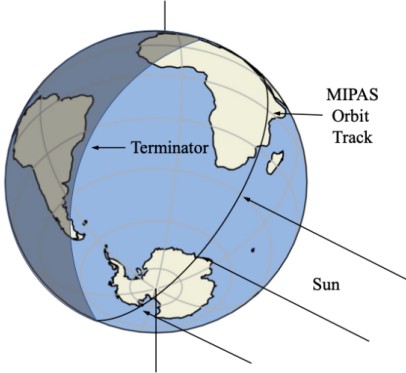

79

80 Figure 1. Schematic representation of the MIPAS orbit at high latitude in December showing the ascending (dayside) and

81 descending (nightside) portions of the orbit and the terminator.

82 **2.2 Model Calculations**

83 The Whole Atmosphere Community Climate Model version 6 (WACCM6) is used in this study. WACCM6 is a component of the

84 Community Earth System Model version 2 (CESM2; Gettelman et al., 2019; Danabasoglu et al., 2020). The horizontal resolution

85 is 1.9° latitude × 2.5° longitude and the with 88 vertical levels up to about 140 km, with the altitude resolution increasing from 0.1

86 km near the surface to 1.0 km in the upper troposphere–lower stratosphere (UTLS) and 1–2 km in the stratosphere. This work uses

87 the specified dynamics version of WACCM6, where the atmosphere below 50 km is nudged to the Modern-Era Retrospective

88 Analysis for Research and Applications, version 2 (MERRA-2; Gelaro et al., 2017), temperature and wind fields with a relaxation

89 time of 50 h. The chemistry mechanism includes a detailed representation of the middle atmosphere, with a sophisticated suite of

90 gas-phase and heterogeneous chemistry reactions, including the Ox, NOx, HOx, ClOx, and BrOx reaction families. There are ~100

91 chemical species and ~300 chemical reactions. Reaction rates are updated following Jet Propulsion Laboratory (JPL) 2015

92 recommendations (Burkholder et al., 2015). The photolytic approach is based on both inline chemical modules (<200nm) and a

93 lookup table approach (>200-750nm; see Kinnison et al., 2007). The look-up table (LUT) approach uses the Tropospheric

94 Ultraviolet-Visible Radiation Model (TUV4.2; Madronich, 1987; Madronich and Weller, 1990), an advanced radiation transfer

95 model widely used by the scientific community, using the four-stream pseudospherical discrete ordinates option. Model values for

96 December 2009 at the same times and location as the satellite data are selected to compare with the satellite data, and denoted

97 "Model".

98 **2.3 Chemical Equation**

99 NO is assumed to be in a steady state in the sunlit atmosphere at 20-40 km at least for a zenith angles less than 94°, due to its short

100 lifetime. Using the chemistry discussed above, $J_{NO_2}$ can then be expressed as



$$J_{NO_2} = \frac{[NO]}{[NO_2]} \times \left( k_{NO+O_3} \times [O_3] + k_{NO+ClO} \times [ClO] \right) - k_{O+NO_2} \times [O] \qquad (2)$$
Where k is the rate constant, $J_{NO_2}$ is the photodissociation coefficient of $NO_2$, and $[O_3]$ is the concentration of $O_3$.
To obtain the concentration of O, O is assumed to be in a steady state with ozone at 20-40 km. The concentration of O can be
expressed as
$$[O] = \frac{J_{O_3} \times [O_3]}{k_{O+O_2+M} \times [O_2] \times [M]} \qquad (3)$$
It is worth noting that $J_{O_3}$ in the Eq. (3) comes from the model here, which is a limitation of this study. However, in the stratosphere
below about 33 km [O] has a small effect on $J_{NO_2}$ (less than 8.1 percent). ClO can similarly be ignored when altitudes are less than
35 km, where ClO concentrations are small; otherwise using ClO data from MIPAS would introduce large and unnecessary
uncertainty. $HO_2$ and BrO both can react with NO but they are not measured by MIPAS and their contributions to the partitioning
between NO and $NO_2$ are negligibly small at the altitudes considered here. Therefore, we don't consider them in this paper.
## 3 Results and Discussion
### 3.1 NOx Concentration at different altitudes
To better understand the diurnal variation of NOx, concentrations of NO and $NO_2$ from MIPAS and the model are shown in Fig.
2 at different altitudes. The NO and $NO_2$ concentrations from MIPAS and the model show very good overall consistency. The
solar zenith angle of 90 degrees is a clear dividing line, showing that light drives the diurnal variation, and the results are in good
accord with the theory.

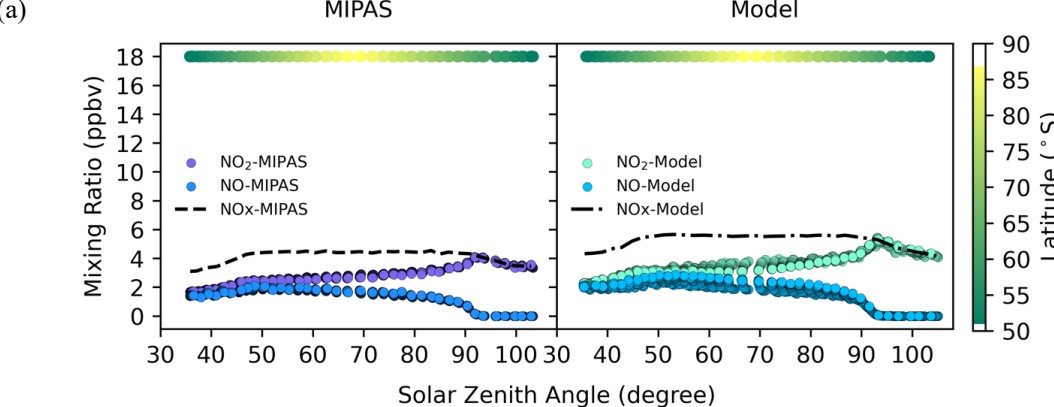





(b)

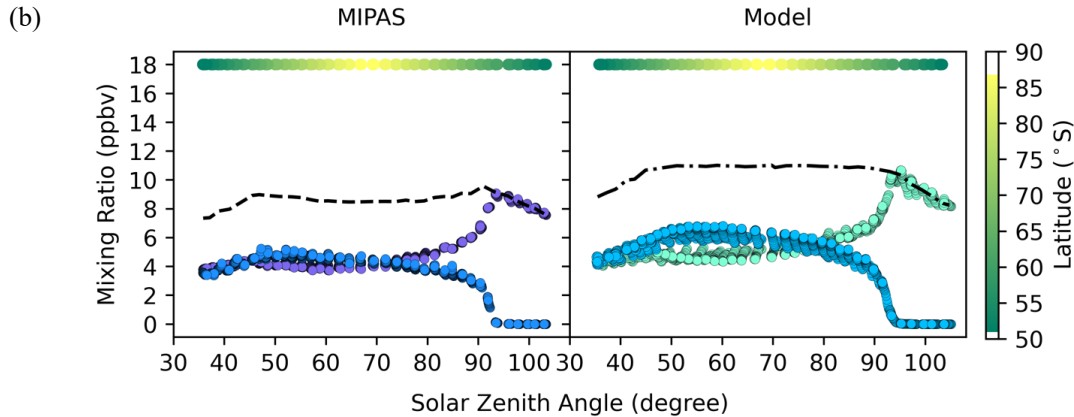


(c)

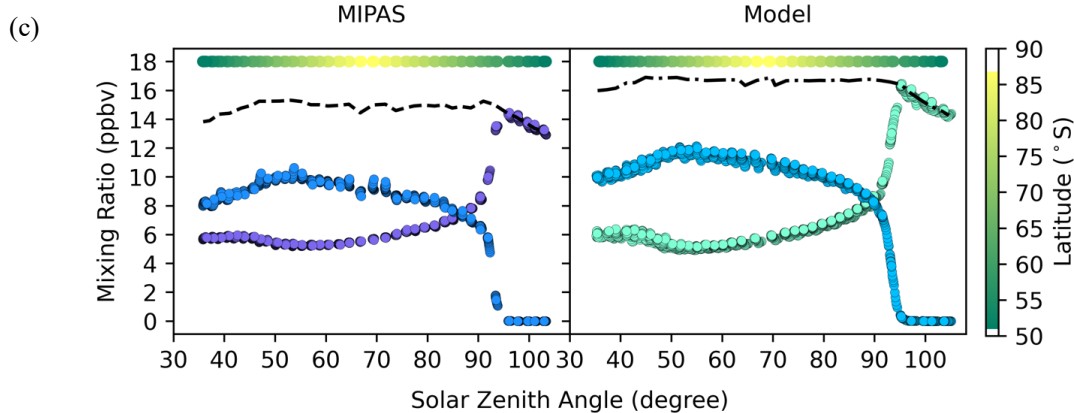


(d)

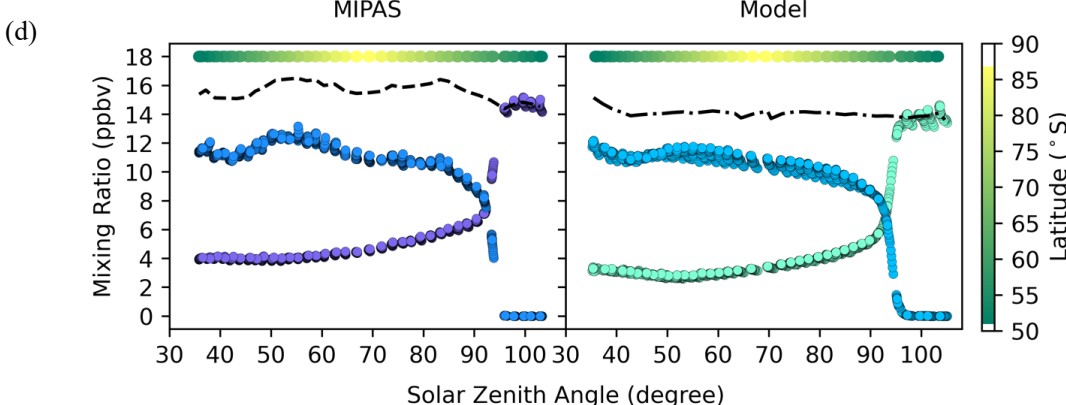


Figure 2. The concentrations of NO, $NO_2$ and $NO_x$ in 50° S-90° S in December 2009 from MIPAS and the model at different
altitudes. (a) 23 km (b) 28 km (c) 33 km (d) 38km. Model values are at the same time and location as the satellite data. The color
bar represents the latitude of the data points at each solar zenith angle. Each point represents the four-day running mean of the
average concentration of multiple daily measurements at two latitude degree intervals.






NO and NO₂ exchange with one another but their sum (NOx) varies relatively little for solar zenith angles less than about 90°. An
increase in NO is matched by a decrease in NO₂ for zenith angles from about 30-50°, and then decreases at larger angles, mainly
reflecting changes in the photolysis rate as the satellite sweeps across the mid-latitudes and polar cap (see below). NO rapidly
disappears when the solar zenith angle exceeds 90 degrees, so the concentrations of NO and NO₂ change dramatically during
twilight. NO decreases rapidly and NO₂ increases rapidly. When the solar zenith angle is more than 90 degrees at these altitudes,
NO is completely oxidized to NO₂, so there is no NO and NO₂/NOx is 1. In addition, the concentration of NO₂ decreases when the
solar zenith angle is more than 90 degrees, which indicates the formation of the N₂O₅.
It should also be noted that in Fig. 2, the concentrations of NO and NO₂ also reflect latitude variations, because the data at each
zenith angle come from different latitudes as shown by the color bar at the top of each panel in Fig. 2, but these variations are fairly
small over the summer polar cap and consistent with the model as shown. From 23 km to 33 km, the concentrations of NO and
NO₂ increase with the altitude.

### 3.2  $J_{NO_2}$ at different altitudes and solar zenith angles

Using Eq. (2) above, the $J_{NO_2}$ at different altitudes is shown in Fig. 3 along with the calculated $J_{NO_2}$ values from the model.  The
correlation diagrams show that the values inferred from the satellite observations are in excellent agreement with the model. Fig.
3 shows that from 20-40 km, the $J_{NO_2}$ values at different altitudes are nearly the same. This indicates the weak dependence of $J_{NO_2}$
value on altitude, which was also reported by Madronich et al. (1985). This is because the NO₂ photolysis is largely driven by
wavelengths ranging from 300 nm to 420nm (Madronich et al., 1983). This spectral region is relatively free of atmospheric
absorption, so the flux is nearly the same at different altitudes. When the solar zenith angle is higher than about 90 degrees, the
$J_{NO_2}$ value drops rapidly to 0. To illustrate how different species affect our calculations at some altitudes, the effects of different
gases at 38 km are shown in Fig. S1. The figure shows that O₃, O and ClO are critical to NOx chemistry at 38 km. However, the
concentrations of ClO and O are smaller at altitudes of less than 35 km, and have about 3.6% and less than 12% influence in our
calculations, respectively. Moreover, the satellite data error of ClO becomes large at lower levels, so ClO is not considered here
when the altitude is lower than 35km.







Figure 3. The $J_{NO_2}$ in 50° S-90° S from MIPAS and the model at different altitudes. (a) 23 km (b) 28 km (c) 33 km (d) 38km. Model values are for the same time and location as the satellite data. The color bar represents the latitude of the data points at nearly the same solar zenith angle. Each point in $J_{NO_2}$-Model and $J_{NO_2}$-MIPAS represents the four-day running mean of the average $J_{NO_2}$ of multiple daily measurements at two latitude degree intervals. In the correlation plots, the abscissa is $J_{NO_2}$-MIPAS and the ordinate is the $J_{NO_2}$-Model and the slope of dashed line is 1.

### 3.3 $J_{NO_2}$ at different latitudes

The $J_{NO_2}$ values from the satellite and model at different latitudes are next discussed. The clear relationship between $J_{NO_2}$ and latitude from MIPAS and model is also displayed in Fig. 4, and the close comparison between the two is remarkable. It is obvious that the satellite-inferred $J_{NO_2}$ monotonically increases with latitude from 30° S-70° S, and then decreases at higher latitudes. The $J_{NO_2}$ over the pole is taken at a larger solar zenith angle, which explains its decrease relative to surrounding parts of Antarctica. Fig. 5 displays maps of the detailed distributions of $J_{NO_2}$ from MIPAS and model, which exhibits their excellent consistency and shows a sharp transition between mid-latitudes and the Antarctic continent or regions covered by sea ice.




The sharp transitions in $J_{NO_2}$ values shown in Fig. 5 can only be caused by the large difference in albedo between the ocean and
the Antarctic environs, covered by sea ice, land ice, and snow (Brandt et al., 2005; Shao and Ke, 2015). Albedo has a strong
influence on $J_{NO_2}$ because $NO_2$ is more sensitive than most atmospheric species to the effects of scattering and reflection
(Madronich et al., 1983; Madronich, 1987; Bösch et al., 2001; Laepple et al., 2005). In the high latitude area, the ground is covered
with ice and snow, and the albedo can be as high as 0.9, while in the lower latitudes, the albedo is about 0.1 (Brandt et al., 2005;
Shao and Ke, 2015). Table 1 shows the $J_{NO_2}$ values at different solar zenith angles under different albedos. The results show that
the albedo has a strong influence on the values, especially at low solar zenith angles. Based on Fig. 5, the $J_{NO_2}$ above the continental
ice is greater than that above the Antarctic sea ice, which may be because the fraction of open water within the pack influences the
albedo (Brandt et al., 2005).

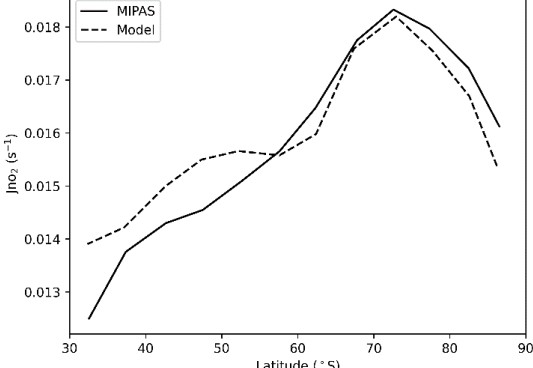


Figure 4. The relationship between $J_{NO_2}$ and latitude from MIPAS and model in 30° S-90° S at 28 km. Model data are for the
same time and location as the satellite data. $J_{NO_2}$ is examined wherever the solar zenith angle is less than 70 degrees and
averaged every five degrees of latitude.

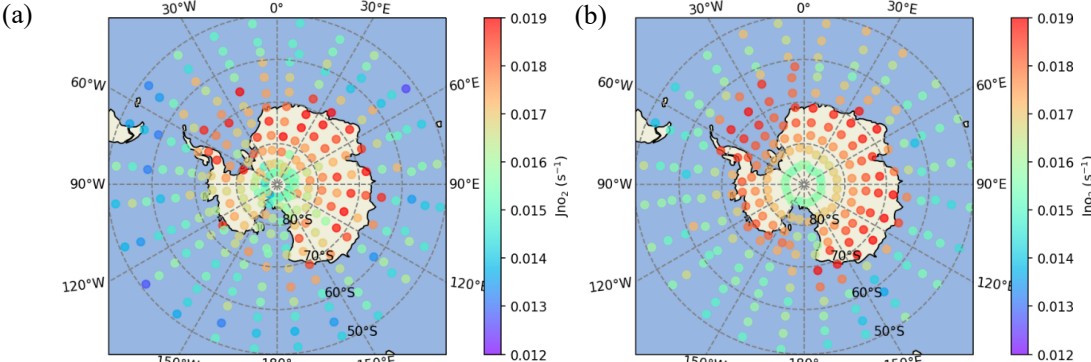




(c)

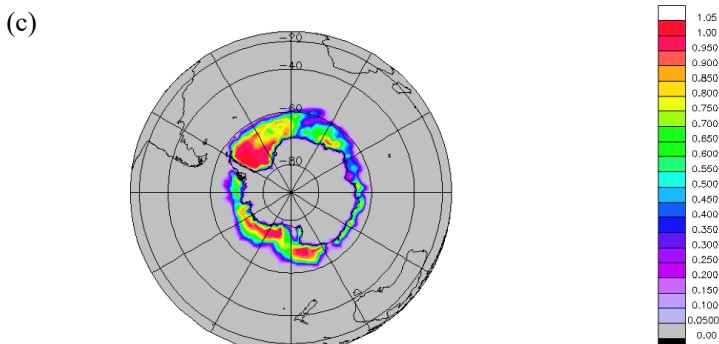


Figure 5. The mapping of $J_{NO_2}$ in 50° S-90° S at 28 km from (a) MIPAS and (b) model. (c) The distribution of the sea ice extent
in December, 2009 in Antarctica and its albedo, from the model. Model $J_{NO_2}$ data are for the same time and location as the
satellite data. $J_{NO_2}$ is shown wherever the solar zenith angle is less than 70 degrees and averaged every 3.33° latitude ×15°
longitude.

Table 1. $J_{NO_2}$ at different solar zenith angles under different albedos ($\alpha$)

| Solar zenith angle | $J_{NO_2}$ ($\alpha = 0.1$) | $J_{NO_2}$ ($\alpha = 0.9$) | $J_{NO_2}$ ($\alpha = 0.9$)/$J_{NO_2}$ ($\alpha = 0.1$) |
|---|---|---|---|
| 0 | 1.30 | 2.21 | 1.70 |
| 10 | 1.30 | 2.19 | 1.69 |
| 20 | 1.29 | 2.14 | 1.65 |
| 30 | 1.28 | 2.05 | 1.59 |
| 40 | 1.27 | 1.92 | 1.52 |
| 50 | 1.24 | 1.77 | 1.42 |
| 60 | 1.20 | 1.58 | 1.31 |
| 70 | 1.14 | 1.36 | 1.20 |
| 80 | 1.01 | 1.10 | 1.09 |
| 90 | 0.664 | 0.673 | 1.01 |


**4   Conclusions**
The diurnal variations of NOx species and the resulting $J_{NO_2}$ from about 50° S-90° S in December in 20-40 km have been evaluated
based on MIPAS data. Light has a strong impact on the diurnal variations. NO and $NO_2$ are in steady state in the daytime and their
sum is almost constant.



The calculated $J_{NO_2}$ remarkably consistent with the model results, and the $J_{NO_2}$ value decreases as the solar zenith angle increases.
The $J_{NO_2}$ value drops rapidly to 0 at the solar zenith angle of about 90 degrees. Moreover, the weak dependence of the $J_{NO_2}$ value
on altitude in this region is evident.
The results from the satellite and the model both indicate that $J_{NO_2}$ increases with latitude, which can be attributed to more reflected
light from ice and snow surfaces with high albedo. In summary, this work presents a new method for obtaining accurate $J_{NO_2}$
values based on satellite data. Further, this method can be extended to other photodissociation coefficients, paving the way for
further tests of global photodissociation coefficients data based on satellites.

**Code and data availability.** The data and code are available at https://doi.org/10.5281/zenodo.7764756.
**Supplement.**
**Author contributions.** S.S. designed the study. J. G. analyzed the data and produced the figures. S. M. and D. K. run the models
and contributed significantly to the interpretation of findings. J.G. wrote the manuscript, with comments from all authors.
**Competing interests.** The contact author has declared that none of the authors has any competing interests.
**Disclaimer.** Publisher's note: Copernicus Publications remains neutral with regard to jurisdictional claims in published maps and
institutional affiliations.
**Acknowledgements.** Doug Kinnison was funded in part by NASA (grant no. 80NSSC19K0952). SS acknowledges support as the
Martin Professor of environmental studies at MIT, while JG appreciates an MIT presidential fellowship. SM acknowledges partial
support by the US Department of Agriculture (USDA) UV-B Monitoring and Research Program, Colorado State University, under
USDA National Institute of Food and Agriculture Grant 2019-34263-30552; 2022-34263-38472. The CESM project is supported
by the National Science Foundation and the Office of Science (BER) of the U.S. Department of Energy. We gratefully acknowledge
high-performance computing support from Cheyenne (doi:10.5065/D6RX99HX) provided by NCAR's Computational and
Information Systems Laboratory, sponsored by the National Science Foundation. We thank the Institute of Meteorology and
Climate Research - Atmospheric Trace Gases and Remote Sensing, and Dr. Michael Kiefer and Dr. Gabriele Stiller for MIPAS
data.

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
