# Peer review of "Inferring the Photolysis Rate of NO2 in the Stratosphere 1 Based on Satellite Observations"

_EGUsphere, 2023_

## Author Comment (AC1)

Response to the Referee #1:

Referee' comments (black) and Author Responses (blue):

(The changes in the paper are underlined in this response.)

This Paper on NO2 photolysis reports an interesting method and its successful application. The topic fits well in with the scope of ACP. The paper is well structured and well written. It is concise, i.e., the authors do not waste pages for unnecessary information but; still, as far as I can judge, all necessary information is there, except for information on uncertainties. I like very much how the authors put their work in the context of existing work. Scientifically, the method applied appears sound to me. The study may not be world-shattering because largely existing knowledge is confirmed, but the study increases confidence both in the modelling of J values and the measurements used.

That said, I must say that I am somewhat disappointed that the authors did not try to estimate the uncertainty of their inferred photolysis rates. In order to judge if this study adds incremental knowledge only or is a major step ahead, it would be necessary to contrast the estimated error of the inferred J-values with the estimated uncertainty of the modelled J values that are based on pre-exixting knowledge. I concede that a rigorous quantitative error assessment might be out of reach because the uncertainties of some ingoing quantities are not quite clear. But even a rough-and-ready uncertainty estimate would be far better than nothing. Or the authors could tackle this issue from the other side: They could estimate how accurate NO and NO2 measurements must be to allow a reasonable inference of J(NO2).

Thank you for the comments. We agree with your insights regarding the significance of uncertainty in this study. We therefore added the uncertainty of $J_{NO_2}$ in Figure 3 and Figure S1, along with appropriate descriptions and discussions about uncertainty in the paper.

The uncertainty is based on Equation (2) and (3):

$$J_{NO_2} = \frac{[NO]}{[NO_2]} \times \left(k_{NO+O_3} \times [O_3] + k_{NO+ClO} \times [ClO]\right) - k_{O+NO_2} \times [O]$$

$$[O] = \frac{J_{O_3} \times [O_3]}{k_{O+O_2+M} \times [O_2] \times [M]}$$

We considered the uncertainties of [NO], [NO2], [O3], [ClO], $k_{NO+O_3}$, $k_{NO+ClO}$, $k_{O+NO_2}$ and $k_{O+O_2+M}$ based on the available uncertainty data in the literature. Here we also appreciate your help and suggestions on correction of IMK/IAA V8 paper cited (see below). In the calculation of the uncertainty, we considered the accuracies of different species because each point in Figure 3 is the average of several hundred data points and random error is much smaller than system error (accuracy).

The accuracies of different species and their sources were described in this paper as follows:

In this paper, we used the NO, NO2, O3, ClO, temperature and pressure data from V8 MIPAS retrievals performed with the IMK/IAA level 2 processor. The retrieval of temperature was

reported by Kiefer et al. (2021). For NO retrieval, the method considered the populations of excited NO states (Funke et al., 2005). This implies that photolysis of $NO_2$ is included in the retrieval priors. However, retrieved NO is only weakly dependent on prior knowledge of $J_{NO_2}$ values (10-15%). In our calculations, according to Eq. (2) and (3), NO, $NO_2$ and $O_3$ play comparable roles in calculation of $J_{NO_2}$, reducing the impact of prior knowledge on the final results. Therefore, prior knowledge of $J_{NO_2}$ will have a small effect on our findings as long as prior knowledge of $J_{NO_2}$ is not completely incorrect. The NO retrieval was documented by Funke et al. (2023). These authors reported an accuracy of 8-15% for altitudes of 20 to 40 km. Regarding $O_3$, Kiefer et al (2023) reported an accuracy of 3-8% in the altitude region of interest. The retrievals of $NO_2$ and ClO were described by Funke et al. (2005) and von Clarmann et al. (2009), respectively, with accuracies of 0.2-0.8 ppbv for $NO_2$ and total error of more than 35% for ClO. However, please note that these papers refer to older data versions. Accuracy estimates for V8 ClO and $NO_2$ are not yet available but the values quoted here were used as a rough guideline.

The reaction rate constants of different species and their total error were described in this paper as following:

The $k_{NO+O_3}$, $k_{NO+ClO}$, $k_{O+NO_2}$ and their uncertainties are from JPL (Burkholder et al., 2015), and $k_{O+O_2+M}$ and its uncertainty are from International Union of Pure and Applied Chemistry (IUPAC; Atkinson et al., 2004).

Figure 3 after adding error bars is as follows:

(a)

[Figure]

(b)

[Figure]

[Figure]

Figure 3. The $J_{NO_2}$ in 50° S-90° S from MIPAS and the model at different altitudes. (a) 23 km (b) 28 km (c) 33km (d) 38km. The color bar represents the latitude of the data points at each solar zenith angle. In the correlation plots, the abscissa is $J_{NO_2}$-MIPAS and the ordinate is the $J_{NO_2}$-Model and the slope of dashed line is 1. To ensure clear visual distinction for each point, black outlines are applied around them.

The uncertainties are about 20% for all altitudes from 20-40 km. There is no dominant term for uncertainty. [NO], [NO2], [O3], [ClO], $k_{NO+O_3}$, $k_{NO+ClO}$, $k_{O+NO_2}$ and $k_{O+O_2+M}$ all import errors that cannot be ignored. Compared with $J_{NO_2}$ at other altitudes, the uncertainties at 38 km are bigger. This is because at 38 km, we need to consider ClO in calculation, which is associated with large error.

Another issue I came across finally turned out to be much less dramatic than I first was afraid of. The MIPAS retrieval of NO uses a model that calculates the populations of the excited NO states. This model takes photolysis of NO2 into account, and uses the respective J values (See Funke et al., , their R4). This suggests that there is the risk of a logical circle when MIPAS NO is used to infer the NO2 photolysis rates. To clarify this, I contacted B. Funke, who gave the all-clear: Chemical excitation accounts only for 10 to 15% in the stratosphere, the rest is thermal. Thus, retrieved NO is only weakly to moderately dependent on the assumed J(NO2) values, and the use of MIPAS NO to confirm the modelled J(NO2) does not lead to a logical circle. However, in cases where the modelled J is totally wrong, the inference method presented by the authors might, strictly

speaking, require an iterative approach where the revised J(NO2) is used for an improved MIPAS NO retrieval, and so forth. Since the authors confirm the modelled J(NO2) values, this is not an issue, but the method presented appears somehow contingent upon reasonable prior knowledge of J(NO2) to me. I wonder if it wouldn't be adequate to mention this issue in the paper. By the way: within the context of an unpublished sanity check of the MIPAS retrieval, B. Funke also inferred J(NO2) from MIPAS data. His results were consistent with the results of the paper under discussion. This increases confidence in the results presented.

Thank you and Dr. Funke for providing the quantitative influence of NO chemical excitation in the stratosphere, which helps us a lot and increases confidence of the result. As you mentioned, the $J_{NO_2}$ was included in the NO retrieval process, and our results are consistent with $J_{NO_2}$ in the model, eliminating the need for iterative calculations of $J_{NO_2}$. We note that retrieved NO is only weakly dependent on prior knowledge of $J_{NO_2}$ values (10-15%). In our calculations, according to Eq. 2 and 3, NO, $NO_2$ and $O_3$ play comparable roles in calculation of $J_{NO_2}$, further reducing the impact of prior knowledge on the final results. Consequently, prior knowledge of $J_{NO_2}$ will have a small effect on our findings. Considering its minor impact and the focus of the paper, we only mention it in the paper without discussing it in detail. We also want to give more information by citing the indicated paper so readers can get more information about the NO retrieval if they are interested in it. We mentioned it and cited the paper as follows:

For NO retrieval, the method considered the populations of excited NO states (Funke et al., 2005). This implies that photolysis of $NO_2$ is included in the retrieval priors. However, retrieved NO is only weakly dependent on prior knowledge of $J_{NO_2}$ values (10-15%). In our calculations, according to Eq. (2) and (3), NO, $NO_2$ and $O_3$ play comparable roles in calculation of $J_{NO_2}$, reducing the impact of prior knowledge on the final results. Therefore, prior knowledge of $J_{NO_2}$ will have a small effect on our findings as long as prior knowledge of $J_{NO_2}$ is not completely incorrect.

Thanks for the information, the consistency of our results and Dr. Funke's should increase confidence both in the modelling of J values and the measurements used.

A further concern is that in some cases the references are misleading because they do not always refer to the data actiually used. In summary, I recommend publication of the manuscript after fixing the issues listed below.

Thanks for the comment and references. We have corrected and updated all references.

I22: strictly speaking, the sunlight entering the stratosphere depends on all absorbing species above. And, even more strictly speaking, as soon as surface-reflected sunlight plays a role, also concentrations of trace gases below may play a role. I suggest to weaken the statement accordingly, e.g. "... as well as the distributions of absorbing species, particularly the overhead concentrations of oxygen and ozone,..."

Thanks for the suggestion. We corrected the sentence as follows:

The properties of sunlight entering the stratosphere, including light intensity and its energy distribution, depend on the solar zenith angle, as well as distributions of absorbing species.

I31: I would prefer Greek letter "nu" instead of "v" for frequency (This may simply be an issue with the font chosen that does not allow to distinguish between nu and v)

Thanks. Corrected.

I58: MIPAS does not measure backscattered radiance but infrared emission of the atmosphere. Just delete "backscatter". The fact that MIPAS measures limb emission is mentioned in the following sentence anyway.

Thank you for the comment. Deleted.

I69: For the species under consideration, the vertical resolution reported here seems a bit optimistic to me. Funke (ACP 16 8667-8693, 2016) report a vertical resolution of 3-6 km for the sunlit stratosphere for NO; Funke et al., ACP 16, 8667-8693, 2016 report a vertical resolution of MIPAS NOy of 4-6 km. von Clarmann et al. (AMT 2, 1-17, 2009) report a vertical resolution of MIPAS ClO of 3.3-12.8 km for the altitude range 20-40 km. For ozone, the vertical resolution estimate seems realistic to me. However, the numbers quoted above refer to older data versions; I understand that version 8 data are used here; values might differ from those quoted above. But I think that the actual vertical resolution is reported along with the mixing ratios in the database and should be used here. Please check the actual values in the database.

Thanks. We have checked the vertical resolution in the database, which is consistent with your information. For $O_3$, the vertical resolution is about 3 km. The vertical resolutions of NO and $NO_2$ are both 3-5 km for the altitude range 20-40 km. The vertical resolution of ClO is 3-8 km for the data we use. Meanwhile, H. Fischer et al. (2008) mentioned the vertical spacing in the stratosphere is 3-8 km. Therefore, for the sake of rigor, we changed the 3 km to 3-8 km. (The change in the paper see below)

I69: The horizontal resolution is 30 km across track. Along-track it is much coarser; for most species it is limited by the along-track sampling, which is appr. 500(400) km until(after) 2004.

Thanks. We updated the description of vertical resolution and horizontal resolution as follows:

The vertical resolution of MIPAS is approximately 3-8 km in the stratosphere, and the horizontal resolution is 30 km across track, about 500 km for along-track until 2004 (400 km after 2004).

I76: Since there exist multiple MIPAS data sets, it may be clearer to write "V8 level2 MIPAS IMK/IAA retrievals". Some of the references refer to older data versions or to another data product not used here.

Thank you. Corrected.

I76/77: V8 NO errors are reported in Funke et al., 2022. This paper is meanwhile accepted for publication. The numbers (5-15% at altitudes between 20 and 40 km) are still valid, but the reference to Sheese at al. is obsolete because it refers to an older data version.

Thanks. The paper cited is updated (Funke et al., 2023). Meanwhile, we checked all preprint papers cited in our paper and updated V8 $O_3$ paper (Kiefer et al. 2023). The reference of Sheese at al. was deleted. The accuracy was also updated (See the uncertainty part above).

I77: The NO2 random uncertainty reported in Wetzel et al. 2007 does not refer to the IMK/IAA data used here but to the ESA data product. Thus this reference is misleading in this context and should be removed. Unfortunately, no error estimates for IMK/IAA V8 NO2 data are available yet. Please see my suggestion below. V8 ozone uncertainties are reported by Kiefer et al. (2022). Thus, the reference to Laeng et al., which refers to an older IMK/IAA data version, is obsolete.

Thanks. Deleted Wetzel et al. (2007) and Laeng et al. (2014). Updated the uncertainties according to your suggestion. (See the uncertainty part above)

I78: Also von Clarmann et al. (2009), referenced here for ClO uncertainties, refers to an older data version. Unfortunately, also for IMK/IAA V8 ClO no journal paper exists yet.

Thanks for the information. The uncertainties of all species were updated according to your suggestion. (See the uncertainty part above)

I76-78: If the authors still want to acknowledge the work referenced (except for the obsolete references), I suggest something like "In this paper, NO, NO2, O3, ClO, temperature and pressure data from V8 MIPAS retrievals performed with the IMK/IAA level 2 processor were used. The retrieval of pressure and temperature is reported by Kiefer et al. (2021). The NO retrieval is documented by Funke et al., 2022. These athors report a precision of 5-15% for altitudes of 20 to 40 km. For O3, Kiefer et al (2022) report a precision of 2-5% in the altitude region of interest. The retrieval of NO2 and ClO is described in Funke et al. (doi:101029/2004JD005225, 2005) and von Clarmann et al. (2009), respectively, with precisions due to measurement noise of 0.2 - 0.3 ppbv for NO2 and more than 35 % for ClO but these papers refer to older data versions. Precision

estimates for V8 ClO and NO2 are not yet available but the values quoted here can be used as a rough guideline." Since no quantitative use of the error estimates is made, all this does not harm the conclusions of the paper. But it should be avoided that the readers are misguided, thus my fussy comments in this context.

Thank you for the valuable suggestion and references. We appreciate your efforts to prevent any potential confusion. The detailed uncertainty calculation was shown in the uncertainty part above. The satellite uncertainties we used are as follows:

In this paper, we used the NO, $NO_2$, $O_3$, ClO, temperature and pressure data from V8 MIPAS retrievals performed with the IMK/IAA level 2 processor. The retrieval of temperature was reported by Kiefer et al. (2021). For NO retrieval, the method considered the populations of excited NO states (Funke et al., 2005). This implies that photolysis of $NO_2$ is included in the retrieval priors. However, retrieved NO is only weakly dependent on prior knowledge of $J_{NO_2}$ values (10-15%). In our calculations, according to Eq. (2) and (3), NO, $NO_2$ and $O_3$ play comparable roles in calculation of $J_{NO_2}$, reducing the impact of prior knowledge on the final results. Therefore, prior knowledge of $J_{NO_2}$ will have a small effect on our findings as long as prior knowledge of $J_{NO_2}$ is not completely incorrect. The NO retrieval was documented by Funke et al. (2023). These authors reported an accuracy of 8-15% for altitudes of 20 to 40 km. Regarding $O_3$, Kiefer et al (2023) reported an accuracy of 3-8% in the altitude region of interest. The retrievals of $NO_2$ and ClO were described by Funke et al. (2005) and von Clarmann et al. (2009), respectively, with accuracies of 0.2-0.8 ppbv for $NO_2$ and total error of more than 35% for ClO. However, please note that these papers refer to older data versions. Accuracy estimates for V8 ClO and $NO_2$ are not yet available but the values quoted here were used as a rough guideline.

I76: A general remark on the use of error estimates: The authors refer to the precision. Isn't the accuracy or the total estimated error more relevant in the given context? I assume that any bias will affect the inferred J-values, while precision (random error) might not be too much of an issue as long as enough measurements are available.

We completely agree that accuracy is more important for uncertainty. Each J-value point is the average of all data from the same latitude for four consecutive days, including several hundred data points. As the result, the total random error is very small through several hundred data average and random error (precision) can be ignored relative to system error (accuracy). Therefore, we only consider the systematic error (accuracy) when calculating uncertainty of J-value. We deleted the precision part and gave the accuracy information instead. The detailed accuracy information was exhibited in uncertainty part above.

I101: I have not quite understood if Eq. (2) is evaluated datapoint by datapoint or if any kind of averaging or regression is involved. If the latter is the case, the precision of the measurements would even be less relevant compared to any possible systematic error contributions.

Yes. In Eq. 2, each J-value point represents the average of all data from the same latitude for four consecutive days, which involves several hundred data points. As you pointed out, the precision of the measurements is less relevant compared to systematic error contributions. Therefore, we deleted the precision part and provide the accuracy information instead.

I99-110: The retrieval of NO by Funke et al. involves a non-LTE model, which calculates the populations of the excited states of NO. This model involves NO2 photolysis and uses TUV photolysis rates. Please see my general comments above on this issue.

Thanks. We responded to this comment following your general comments.

I169/170: A short explanation why NO2 photolysis is more albedo-dependent than that of other species would be nice. I suspect that this is because, as stated above, the atmosphere is quite transparent at frequencies relevant to NO2 photolysis, thus enough backscattered photons survive the long path through the atmosphere to Earth's surface and back to the air volume under assessment.

Yes, you are right. As stated, $NO_2$ photolysis is largely driven by wavelengths ranging from 300 nm to 420 nm (Madronich et al., 1983). This spectral region is relatively free of atmospheric absorption, while reflections and Rayleigh scattering redistribute much of the incoming sunlight. The $NO_2$ photolysis is more albedo-dependent because of the optical properties. As a result, enough backscattered photons can survive in the long path through the transparent atmosphere to Earth's surface and back to the stratosphere. Furthermore, the significance of albedo outweighs that of Rayleigh scattering. Madronich et al. (1983) reported if only direct sunlight were effective, the maximum value of $J_{NO_2}$ is less than $5 \times 10^{-3} s^{-1}$ for the isotropic model. If diffuse light resulting from multiple scattering is considered, $J_{NO_2}$ can increase to about $7 \times 10^{-3} s^{-1}$. In addition, if all light reaching the ground is reflected isotropically, $J_{NO_2}$ values are as high as $2.8 \times 10^{-2} s^{-1}$. These results also show the importance of albedo for $J_{NO_2}$.

We added the discussion of albedo-dependence in the paper as follows:

Albedo has a strong influence on $J_{NO_2}$ because $NO_2$ is more sensitive than most atmospheric species to the effects of scattering and reflection (Madronich et al., 1983; Madronich, 1987; Bösch et al., 2001; Laepple et al., 2005). This is because the atmosphere exhibits considerable transparency at frequencies relevant to $NO_2$ photolysis, allowing a large number of photons to persist throughout the long atmospheric path, reaching Earth's surface and eventually returning to the stratosphere.

I200: I think that this statement might be a bit too strong or too general. If no J(NO2) data are available, then there is no reliable NO-retrieval from infrared emission measurements (see my discussion of this issue in the general part of this review). I would prefer some slightly weaker

wording here. What the authors certainly can claim is that they present a new method to validate the modelling of J(NO2).

Thanks. We agree that the prior knowledge from model is required to our result although the influence of the prior knowledge from model is small (as stated in the response above).

Therefore, we have slightly adjusted the wording regarding the data source, making it clear that not all data is based on satellite:

In summary, this work presents a new method for obtaining accurate $J_{NO_2}$ values mainly based on satellite data.

Conclusion: You may also wish to add a short remark that the nice agreement between modelled photolysis rates and those inferred from the measurements also increases confidence in the measurements used. Does the excellet fit between inferred and modelled J-values imply that the estimates of measurement errors of NO and NO2 are overly conservative?

We agree that our results can yield a conclusion about MIPAS measurement error, not only for NO and $NO_2$, but also for $O_3$. Our results are consistent with $J_{NO_2}$ from model perfectly, and the deviation between our results and model is much smaller than the 1-$\sigma$ uncertainty. Therefore, we think the estimates of measurement errors of MIPAS are conservative, especially for NO and $NO_2$.

Therefore, the conservative estimates of measurement errors of MIPAS were discussed in the part of the discussion dealing with uncertainty.

The uncertainty is also shown in Fig. 3, and the model $J_{NO_2}$ is within the error bar. The deviation between the results and model is significantly smaller than the 1-$\sigma$ uncertainty, implying that the estimates of measurement errors of MIPAS may be conservative.

---

## Author Comment (AC2)

Response to the Referee #2:

Referee' comments (black) and Author Responses (blue):

(The changes in the paper are underlined in this response.)

The authors present a new method to infer the stratospheric photolysis rate of $NO_2$ using satellite (MIPAS) measurements. The photolysis rate coefficient determines the diurnal variation of NOx photochemistry. The results agree well with model predictions. This work provides the first observations-based validation of the role of albedo in driving polar photochemistry.

The scientific questions addressed by the paper are certainly within the scope of ACP. The title clearly reflects the contents of the paper, the abstract provides a concise and complete summary of the work, the authors provide proper credit to related work by other groups and clearly indicate their new contribution. The presentation is generally well structured and clear, and the language and mathematical notation is adequate.

My main concern is that the description of the approach is not sufficiently complete to allow their reproduction by others. For example, how do the authors collocate the model horizontal and vertical grid with those of the measurements? It is stated that "Model values for December 2009 at the same times and location as the satellite data are selected to compare with the satellite data", but what are the actual spatial and temporal collocation criteria? Further, is any kind of interpolation (temporal, horizontal and/or vertical) done subsequently to match the grids up? This is critical information that is missing in the paper. Are aerosols or clouds considered in the comparisons? This is relevant because the authors use a four-stream radiative transfer model, which may not be accurate enough (especially in the UV/Visible spectral regions) when aerosols or clouds are present. The retrieval accuracies may also degrade in these scenarios.

Thank you for the comments. We have improved the description of the methods.

The model's satellite profile algorithm outputs constituents or rates (e.g., NO concentration, or $Jno_2$) at the nearest available latitude, longitude, and local time to those of each observation. The model resolution for this study is ~2 degrees in the horizontal; therefore, the spatial resolution accuracy would be within +/- 100km in each horizontal direction (+/- 140km along the diagonal direction). The vertical resolution is dependent on the model vertical resolution. The model chemistry time step is 30 minutes; therefore, the temporal resolution is +/- 15 minutes.

The photolysis routine is based on a lookup table approach (Kinnison et al., 2007). This approach does not include clouds or aerosols in the radiative transfer. However, there is a cloud correction factor applied to the total photolysis rate based on Chang et al., 1987 (equations 12-14). This photolysis cloud correction approach is discussed in Brasseur et al., 1998. There is no correction factor included for aerosols. In this paper, we didn't exclude the data when aerosols and clouds are present.

Chang, J. S., R. A. Brost, I. S. A. Isaksen, S. Madronich, P. Middleton, W. R. Stockwell, and C. J. Walcek, Three-Dimensional Eulerian Acid Deposition Model: Physical Concepts and Formulation, J. Geophys. Res., vol. 92, NO. D12, 14,681-14,700, 1987.

Brasseur, G. P., D. A., Hauglustaine, S. Walters, P. J. Rasch, J.-F. Muller, C. Granier, and X. X. Tie, MOZART, a global chemical transport model for ozone and related chemical tracer: Model description, J. Geophys. Res., vol. 103, NO. D21, 28,265-28,290, 1998.

Kinnison, D. E., G. P. Brasseur, S. Walters, R. R. Garcia, F. Sassi, B. A. Boville, D. Marsh, L. Harvey, C. Randall, W. Randel, J. F. Lamarque, L. K. Emmons, P. Hess, J. Orlando, J. Tyndall, and L. Pan, Sensitivity of chemical tracers to meteorological parameters in the MOZART-3 chemical transport model, J. Geophys. Res., 112, D20302, doi:10.1029/2006JD007879, 2007.

Therefore, we added the description of the satellite profile algorithm as follows:

Model values for December 2009 at the same times and locations as the satellite data are selected by the satellite profile algorithm to compare with the satellite data, and denoted "Model". The satellite profile algorithm outputs constituents (e.g., $J_{NO_2}$ and NOx concentrations) at the nearest latitude, longitude, and local time to the observation.

Furthermore, to ensure better clarity and accuracy in this paper, we have expanded the descriptions of data processing and data sources.

Data processing: The data from the satellite was averaged daily and zonally (Because the specific latitudes of the satellite data vary somewhat from one orbit to another, we bin the data using a two-degree interval). Then we calculate the four-day running mean, which is shown in Fig. 2 and Fig. 3.

Reaction rate constants sources: The $k_{NO+O_3}$, $k_{NO+ClO}$, $k_{O+NO_2}$ and their uncertainties are from JPL (Burkholder et al., 2015), and $k_{O+O_2+M}$ and its uncertainty are from International Union of Pure and Applied Chemistry (IUPAC; Atkinson et al., 2004).

Figure descriptions: To ensure clear visual distinction for each point, black outlines are applied around them.

The uncertainties were also discussed in more detail, please see uncertainty part below.

This is relevant because the authors use a four-stream radiative transfer model, which may not be accurate enough (especially in the UV/Visible spectral regions) when aerosols or clouds are present. The retrieval accuracies may also degrade in these scenarios.

Regarding your inquiry about the adequacy of the 4-stream discrete ordinates, we conducted a brief sensitivity study to validate the accuracy of the four-stream radiative transfer model. In this test, we compare the performance of 2, 4, 8, 16 stream radiative transfer, under both high and low albedo scenarios. The Delta-Eddington 2 stream model is the fastest model, and the Discrete Ordinates 4 stream model is used in WACCM. We also use Discrete Ordinates 8 stream model and Discrete Ordinates 16 stream model, which is the most accurate and serves as the standard for this test.

Subsequently, we calculated the $Jno_2$ at 30 km under clear sky conditions without the presence of aerosols or clouds using these different radiative transfer models. The $Jno_2$ and the difference with 16-stream model, under surface albedo conditions of 0.1 and 0.9, are presented in Table 1 and Table 2, respectively.

Table 1. The $Jno_2$ and the errors of 2, 4, 8, 16 stream models under the surface albedo of 0.1

| sza | 2-stream | 4-stream | 8-stream | 16-stream | Err2 (%) | Err4 (%) | Err8 (%) |
|---|---|---|---|---|---|---|---|
| 0 | 1.26E-02 | 1.30E-02 | 1.30E-02 | 1.30E-02 | -3.07 | -0.31 | -0.23 |
| 10 | 1.26E-02 | 1.30E-02 | 1.30E-02 | 1.30E-02 | -3.15 | -0.23 | -0.15 |
| 20 | 1.25E-02 | 1.29E-02 | 1.29E-02 | 1.30E-02 | -3.63 | -0.31 | -0.23 |
| 30 | 1.23E-02 | 1.28E-02 | 1.28E-02 | 1.29E-02 | -4.20 | -0.31 | -0.23 |
| 40 | 1.21E-02 | 1.27E-02 | 1.27E-02 | 1.27E-02 | -5.03 | -0.39 | -0.24 |
| 50 | 1.18E-02 | 1.24E-02 | 1.25E-02 | 1.25E-02 | -5.92 | -0.40 | -0.24 |
| 60 | 1.13E-02 | 1.20E-02 | 1.21E-02 | 1.21E-02 | -6.77 | -0.58 | -0.25 |
| 70 | 1.06E-02 | 1.14E-02 | 1.14E-02 | 1.15E-02 | -7.41 | -0.87 | -0.35 |
| 80 | 9.49E-03 | 1.01E-02 | 1.02E-02 | 1.03E-02 | -7.48 | -1.46 | -0.29 |
| 90 | 6.49E-03 | 6.64E-03 | 6.73E-03 | 6.78E-03 | -4.34 | -2.11 | -0.72 |

Note that sza is solar zenith angle and the error is the difference with 16-stream model, which serves as the standard for this test.

Table 2. The $Jno_2$ and the errors of 2, 4, 8, 16 stream models under the surface albedo of 0.9

| sza | 2-stream | 4-stream | 8-stream | 16-stream | Err2 (%) | Err4(%) | Err8(%) |
|---|---|---|---|---|---|---|---|
| 0 | 2.28E-02 | 2.21E-02 | 2.21E-02 | 2.22E-02 | 3.12 | -0.32 | -0.23 |
| 10 | 2.26E-02 | 2.19E-02 | 2.19E-02 | 2.20E-02 | 2.96 | -0.32 | -0.23 |
| 20 | 2.20E-02 | 2.14E-02 | 2.14E-02 | 2.14E-02 | 2.57 | -0.28 | -0.19 |
| 30 | 2.09E-02 | 2.05E-02 | 2.05E-02 | 2.05E-02 | 1.80 | -0.34 | -0.24 |
| 40 | 1.95E-02 | 1.92E-02 | 1.93E-02 | 1.93E-02 | 0.78 | -0.41 | -0.26 |
| 50 | 1.77E-02 | 1.77E-02 | 1.77E-02 | 1.78E-02 | -0.56 | -0.45 | -0.28 |
| 60 | 1.56E-02 | 1.58E-02 | 1.59E-02 | 1.59E-02 | -2.14 | -0.57 | -0.31 |
| 70 | 1.32E-02 | 1.37E-02 | 1.37E-02 | 1.38E-02 | -3.92 | -0.80 | -0.29 |
| 80 | 1.06E-02 | 1.10E-02 | 1.12E-02 | 1.12E-02 | -5.54 | -1.34 | -0.27 |
| 90 | 6.61E-03 | 6.73E-03 | 6.83E-03 | 6.88E-03 | -3.91 | -2.11 | -0.73 |

WACCM uses the 4 stream discrete ordinates model. Errors in $Jno_2$ at 30 km computed with a 4 streams model, relative to 16 streams model, are less than 1% for sza<75, reaching about 2% at sza=90. The error is always negative, causing a slight underestimation of the true value. Two-stream errors are substantially larger, positive or negative, and can reach 7 or 8 %. In this test, we found that the accuracy of the 4-stream model is nearly equivalent to that of the 16-stream model, indicating that the 4-stream model provides sufficient accuracy for our purposes.

The work does present an "observations-based check on the role of albedo in driving polar photochemistry", but this result alone would only provide an incremental improvement to existing scientific understanding. It would be a lot more revealing if the authors could figure out under what conditions the models do not work so well (scenarios with aerosols and/or clouds?).

In this paper we introduce a novel approach to derive precise $J_{NO_2}$ values based on satellite data. This paper focuses on the methodology, which is totally different from model. In the model, taking TUV model as example, it is based on radiative transfer principles. By accounting for the vertical distribution of atmospheric constituents, the TUV model can calculate the intensity and spectral distribution of UV and visible radiation at different altitudes. Then based on photochemical data (such as from JPL), calculating the $J_{NO_2}$ values. Our satellite method is grounded in the steady state assumption. We calculate the $J_{NO_2}$ values using satellite data through Equations 2 and 3. The key message of our paper is that using two totally different methods, $J_{NO_2}$ calculated by satellite data and $J_{NO_2}$ from the model are consistent with each other, significantly increasing the confidence in the existing $J_{NO_2}$ values. Moreover, future studies could verify other photolysis rates using our method. In addition, an implication of the paper is conceptual: Obtaining photolysis rates through satellite data. The benefits of obtaining photolysis rates through satellite data are significant: We can obtain global uninterrupted data, similar to species concentration data from satellites, which can be used for more refined research. We can also test and confirm the impact of surface albedo, an important factor in atmospheric photolysis and chemistry, tested here with the contrast from the Antarctic to lower latitudes. Indeed, due to the uncertainty of satellites, the temporal and spatial resolutions for obtaining effective $J_{NO_2}$ are currently very low. In this paper, a point represents the average of the data in the same latitude in four days. But with the advancement of satellite sensors and retrieval methods, the temporal and spatial resolution will become smaller and smaller, which supports the further research for better understanding the $J_{NO_2}$.

In summary, our paper not only significantly increases the confidence in the existing $J_{NO_2}$ knowledge, and introduces a method that can be extended to other photolysis rates, but also presents a new promising concept that obtaining photolysis rates through satellite.

This also leads to the issue of uncertainty quantification. There is no mention of error characteristics in the paper. This is critical for satellite-based retrievals. Without knowledge of the retrieval errors, it is very hard to make any evaluations about the quality and/or robustness of the results. For example, the statement that "However, in the stratosphere below about 33 km [O] has a small effect on JNO2 (less than 8.1 percent)" is meaningless unless it is contrasted with errors in JNO2 itself. The authors do report precisions for the various species. These could probably be used to obtain precisions for the photolysis rate estimates.

Thank you for the comments. We agree with your insights regarding the significance of uncertainty in this study. We therefore added uncertainties for $J_{NO2}$ in Figure 3 and Figure S1, along with descriptions and discussions about uncertainty in the paper.

The uncertainty is based on the Equation (2) and (3):

$$J_{NO_2} = \frac{[NO]}{[NO_2]} \times \left(k_{NO+O_3} \times [O_3] + k_{NO+ClO} \times [ClO]\right) - k_{O+NO_2} \times [O]$$

$$[O] = \frac{J_{O_3} \times [O_3]}{k_{O+O_2+M} \times [O_2] \times [M]}$$

We considered the uncertainty of $[NO]$, $[NO_2]$, $[O_3]$, $[ClO]$, $k_{NO+O_3}$, $k_{NO+ClO}$, $k_{O+NO_2}$ and $k_{O+O_2+M}$ based on the available uncertainty of each term as given in the literature. In the calculation of the uncertainty, we considered the accuracies of different species because each point in Figure 3 is the average of several hundred data points and random error is much smaller than systematic error (accuracy).

The accuracies of different species and their sources are now described in this paper as follows:

In this paper, we used the NO, NO_2, O_3, ClO, temperature and pressure data from V8 MIPAS retrievals performed with the IMK/IAA level 2 processor. The retrieval of temperature was reported by Kiefer et al. (2021). For NO retrieval, the method considered the populations of excited NO states (Funke et al., 2005). This implies that photolysis of $NO_2$ is included in the retrieval priors. However, retrieved NO is only weakly dependent on prior knowledge of $J_{NO_2}$ values (10-15%). In our calculations, according to Eq. (2) and (3), NO, NO_2 and O_3 play comparable roles in calculation of $J_{NO_2}$, reducing the impact of prior knowledge on the final results. Therefore, prior knowledge of $J_{NO_2}$ will have a small effect on our findings as long as prior knowledge of $J_{NO_2}$ is not completely incorrect. The NO retrieval was documented by Funke et al. (2023). These authors reported an accuracy of 8-15% for altitudes of 20 to 40 km. Regarding O_3, Kiefer et al (2023) reported an accuracy of 3-8% in the altitude region of interest. The retrievals of NO_2 and ClO were described by Funke et al. (2005) and von Clarmann et al. (2009), respectively, with accuracies of 0.2-0.8 ppbv for NO_2 and total error of more than 35% for ClO. However, please note that these papers refer to older data versions. Accuracy estimates for V8 ClO and NO_2 are not yet available but the values quoted here were used as a rough guideline.

The reaction rate constants of different species and their total errors were described in this paper as follows:

The $k_{NO+O_3}$, $k_{NO+ClO}$, $k_{O+NO_2}$ and their uncertainties are from JPL (Burkholder et al., 2015), and $k_{O+O_2+M}$ and its uncertainty are from International Union of Pure and Applied Chemistry (IUPAC; Atkinson et al., 2004).

Figure 3 after adding error bars is as follows:

[Figure]

Figure 3. The $J_{NO_2}$ in 50° S-90° S from MIPAS and the model at different altitudes. (a) 23 km (b) 28 km (c) 33 km (d) 38km. The color bar represents the latitude of the data points at the same solar zenith angle. In the correlation plots, the abscissa is $J_{NO_2}$-MIPAS and the ordinate is the $J_{NO_2}$-Model and the slope of dashed line is 1. To ensure clear visual distinction for each point, black outlines are applied around them.

The uncertainties are about 20% for all altitudes from 20-40 km. There is no dominant term for uncertainty. [NO], [NO2], [O3], [ClO], $k_{NO+O_3}$, $k_{NO+ClO}$, $k_{O+NO_2}$ and $k_{O+O_2+M}$ all impart errors that cannot be ignored. Compared with Jno2 at other altitudes, the uncertainties at 38 km are bigger. This is because at 38 km, we need to consider ClO in calculation, which is associated with large error.

A few statements need references:

Thanks. Added the references as following.

"However, in the stratosphere below about 33 km [O] has a small effect on JNO2 (less than 8.1 percent)."

However, in the stratosphere below about 33 km [O] has a small effect on $J_{NO_2}$ calculation (less than 8.1 percent) due to its low concentration (Johnston and Podolske, 1978).

"ClO can similarly be ignored when altitudes are less than 35 km, where ClO concentrations are small"

ClO can similarly be ignored when altitudes are less than 35 km, where ClO concentrations are small (Sagawa et al., 2013)

"HO2 and BrO both can react with NO but they are not measured by MIPAS and their contributions to the partitioning between NO and NO2 are negligibly small at the altitudes considered here."

HO2 and BrO both can react with NO but they are not measured by MIPAS and their contributions to the partitioning between NO and NO2 are negligibly small at the altitudes considered here (Del Negro et al., 1999).

Overall, the paper has potential for publication after the changes listed above are made.